# Identification of c.1531C>T Pathogenic Variant in the *CDH1* Gene as a Novel Germline Mutation of Hereditary Diffuse Gastric Cancer

**DOI:** 10.3390/ijms20204980

**Published:** 2019-10-09

**Authors:** Enrique Norero, M. Alejandra Alarcon, Christopher Hakkaart, Tomas de Mayo, Cecilia Mellado, Marcelo Garrido, Gloria Aguayo, Marcela Lagos, Javiera Torres, Alfonso Calvo, Parry Guilford, Alejandro H. Corvalan

**Affiliations:** 1Esophagogastric Surgery Unit, Surgery Department, Hospital Dr. Sotero del Rio, Santiago 8207257, Chile; acalvobelmar@yahoo.es; 2Digestive Surgery Department, Hospital Clinico Universidad Católica de Chile, Pontificia Universidad Católica de Chile, Santiago 8330024, Chile; 3Advanced Center for Chronic Diseases (ACCDiS), Pontificia Universidad Católica de Chile, Santiago 8330024, Chile; mralarco@uc.cl (M.A.A.); tomasdemayo@gmail.com (T.d.M.); acorvalan@uc.cl (A.H.C.); 4Cancer Genetics Laboratory, Department of Biochemistry, University of Otago, Dunedin 9054, New Zealand; christopher.hakkaart@otago.ac.nz (C.H.); parry.guilford@otago.ac.nz (P.G.); 5Faculty of Sciences, School of Medicine Universidad Mayor Santiago Chile, Santiago 8580745, Chile; 6Genetic Unit, Pediatrics Division, Pontificia Universidad Católica de Chile, Santiago 8330024, Chile; mellado.c@gmail.com; 7Hematology and Oncology Department, School of Medicine, Pontificia Universidad Católica de Chile, Santiago 8330024, Chile; mgarrido@med.puc.cl; 8Pathology Department, Hospital Dr. Sotero del Rio, Santiago 8207257, Chile; gloriaaguayo@gmail.com; 9Molecular Biology Laboratory, Pontificia Universidad Católica de Chile, Santiago 8330024, Chile; mlagos@med.puc.cl; 10Pathology Department, Pontificia Universidad Católica de Chile, Santiago 8330024, Chile; javitorresm@yahoo.com

**Keywords:** HDGC, *CDH1*, prophylactic gastrectomy

## Abstract

Germline pathogenic variants in the *CDH1* gene are a well-established cause of hereditary diffuse gastric cancer (HDGC) syndrome. The aim of this study was to characterize *CDH1* mutations associated with HDGC from Chile, a country with one of the highest incidence and mortality rates in the world for gastric cancer (GC). Here, we prospectively include probands with family history/early onset of diffuse-type of GC. The whole coding sequence of the *CDH1* gene was sequenced from genomic DNA in all patients, and a multidisciplinary team managed each family member with a pathogenic sequence variant. Thirty-six cases were included (median age 44 years/male 50%). Twenty-seven (75%) patients had diffuse-type GC at ≤50 years of age and 19 (53%) had first or second-degree family members with a history of HDGC. Two cases (5.5%) carried a non-synonymous germline sequence variant in the *CDH1* gene: (a) The c.88C>A missense variant was found in a family with three diffuse-type GC cases; and (b) c.1531C>T a nonsense pathogenic variant was identified in a 22-year-old proband with no previous family history of HDGC. Of note, six family members carry the same nonsense pathogenic variant. Prophylactic gastrectomy in the proband’s sister revealed stage I signet-ring cell carcinoma. The finding of 1531C>T pathogenic variant in the *CDH1* in proband with no previous family history of HDGC warrants further study to uncover familial clustering of disease in *CDH1* negative patients. This finding may be particularly relevant in high incidence countries, such as the case in this report.

## 1. Introduction

Gastric cancer (GC) is the fifth most common cancer, with more than 1,033,000 new cases every year, and the third leading cause of cancer-related death worldwide [1]. Most GC’s are sporadic, however, approximately 10% of GC cases have a familial clustering [2]. In 1964, Jones described a multigenerational New Zealand Maori family with an extremely high incidence of GC and speculated about a genetic role [3]. In 1998, Guilford made the first association between familial GC and a germline mutation in the *CDH1* gene in three Maori families [4]. In these patients, a clinical entity called hereditary diffuse gastric cancer (HDGC) has been described [5,6]. 

HDGC is an autosomal dominant cancer syndrome primarily characterized by an extreme risk of developing diffuse-type GC from a relatively young age [5,7]. *CDH1* mutations are highly penetrant and carriers have a lifetime risk of 70–80% of developing GC [8,9], with an estimated cumulative risk of GC by age 80 of 67–70% for men and 56–83% for women [10,11]. More than 120 pathogenic *CDH1* variants have been reported across all coding regions of the *CDH1* gene [10,12]. 

The *CDH1* gene is located in chromosome 16q22.1 and consists of 16 exons that encode a 120-kDa protein called E-cadherin. E-cadherin is expressed in epithelial tissues and is responsible for calcium-dependent cell-cell adhesion [13]. When the *CDH1* gene has a pathogenic variant, there is a loss of cell adhesion, and therefore cells acquire an invasive behavior contributing to tumor development [7,14]. *CDH1* is also an important gene in sporadic GC pathogenesis being the second most mutated gene after *TP53* [15]. On the global level, approximately 14–50% of cases that fit the clinical criteria for HDGC carry a pathogenic variant in germline *CDH1* gene [8,9,10,16]. 

In Latin America, only Colombia and Brazil have reported *CDH1* mutations associated with HDGC [17,18,19]. In Chile, with the highest incidence and mortality rates of GC in the region, family clustering in terms of hereditary disease of GC has not been directly addressed.

In this study, we identified the non-synonymous c.1531C>T mutation in the *CDH1* gene as a novel germline mutation of HDGC in the proband case, as well as in five family members. Prophylactic gastrectomy in the proband’s sister revealed stage I signet-ring cell carcinoma.

## 2. Results

As shown in Table 1, 36 probands were included in the study. There were 13 patients with family history of diffuse gastric cancer (DGC) in first or second-degree family members, with one ≤ 50 years; 13 probands had three or more family members with DGC at any age; and 27 patients had a diagnosis of DGC ≤ 50 years. Seventeen patients (47%) had no other family members with a diagnosis of GC, and they were included in the study because they had age ≤ 50 years when they were diagnosed with GC. The mean age at the time of diagnosis was 44 years old (range 22–68). Fourteen (38%) patients were age 39 or younger and there were 18 male patients (50%). No patients had previously been diagnosed with GC. The majority of patients presented with advanced stage disease (III 33% and IV 36%) (Table 1).

There were also other malignant neoplasms such as breast cancer (16%), colorectal cancer (6%) and other tumors (33%) found in the patient families (Table 1). Histologic classification was not available for most tumors in the family members. 

Treatment was gastrectomy with or without neoadjuvant treatment in 26 (72%) patients, palliative chemotherapy in four (11%), non-resective surgery in four (11%), and supportive care in two (5%). At the completion of the study, six patients with stage IV disease had died from GC.

The germline sequence variants identified in the *CDH1* gene in all tested individuals are described in Table 2. Most sequence variants are classified as benign or likely benign. However, in two (5.5%) patients, two non-synonymous sequence variants were identified. 

The first proband, found to carry a missense c.88C>A (p.Pro30Thr, rs139866691) variant, was a 59-year-old female diagnosed with stage III GC of signet-ring cell subtype. Interestingly, she had a family history of HDGC from which her father and grandfather both died at age 65 and 52 years old, respectively (Appendix A). Additionally, one second-degree and one third-degree family member were diagnosed with prostate cancer at 72 years of age and renal cancer at 4 years of age, respectively. According to ClinVar (www.ncbi.nlm.nih.gov/clinvar/variation/127933/), there are conflicting interpretations regarding the pathogenicity of the c.88C>A variant [20]. Therefore, and according to the current international guidelines [8], this variant is currently classified as a variant of unknown significance (VUS) [21,22].

The second proband, belonging to a different family, was found to carry a non-synonymous c.1531C>T (p.Gln511*, rs1131690810) variant (Figure 1). This case, the youngest in our study, was a 22-year-old male diagnosed with stage IV GC, and also the signet-ring cell subtype. The patient received palliative chemotherapy and died 3 months after diagnosis. Although at the time of diagnosis, the proband did not have any known family history of GC, 8 members of the family were genetically screened to determine carrier status of c.1531C>T variant. As shown in Figure 2, five family members were identified as carrying the same pathogenic variant. High-definition and high-magnification endoscopies were performed in four of these carriers without categorical findings. In all cases, random biopsies were taken, and histological evaluation showed chronic gastritis and intestinal metaplasia. 

All family members were referred to a multidisciplinary team and prophylactic total gastrectomy was recommended. The proband’s sister, a 20-year-old female, received a laparoscopic total gastrectomy with D1 lymph node dissection and has had an uneventful recovery. Pathological examination of the gastrectomy specimen revealed five micro-foci of signet ring cell carcinoma located in the upper and middle third of the stomach, with only mucosal (T1a) extension and no lymph node (N0) metastasis (Stage IA disease) (Figure 3).

## 3. Discussion

This prospective genetic study is the first to describe the pathogenic variant c.1531C>T, as reported in ClinVar (www.ncbi.nlm.nih.gov/clinvar/RCV000492683.1/), in association with the HDGC syndrome. It is worth noting that this variant is not reported in the most robust and comprehensive study of HDGC-novel *CDH1* mutations [10], nor it is listed in the Leiden Open Variation Database for *CDH1* (www.lovd.nl/CDH1). The c.1531C>T (p.Gln511*) nonsense variant causes the termination of the E-cadherin protein at the third extracellular domain. Typically, patients with pathogenic *CDH1* variants develop diffuse-type GC between 35 to 40 years of age [8]. However, in the family in which this mutation was identified, the proband as well as the family carrier who underwent prophylactic gastrectomy were diagnosed at early twenties. Thus, this study highlights the relevance of identifying *CDH1* mutation carriers in families that exhibit early onset of disease (≤ 50 years of age at diagnosis). Because of the high penetrance of this gene [5,9,12], the timely and accurate molecular evaluation of this family led to a prophylactic gastrectomy revealing only mucosal involvement by signet-ring cell carcinoma in the proband’s sister, making this a life-saving procedure [23,24]. A systematic review describes that the final histopathology diagnosis of GC was made in 87% of 169 prophylactic gastrectomies [12], supporting gastrectomy as the best recommendation in *CDH1* mutation carrier individuals [8,9]. In the past years, the development of minimally invasive surgery for GC has improved short-term outcomes in total gastrectomy [25,26,27,28,29]. Laparoscopic total gastrectomy in our study and in reports from other groups supports the applicability of this approach and potentially may improve the acceptance of a surgical option by *CDH1* mutation carriers [18,30]. The family described in our study with the c.1531C>T variant has a unique profile, in which one of the youngest family members developed an advanced GC and older family members did not. We do not understand the full mechanisms behind the progression from normal mucosa or small foci of signet ring cells (present in most prophylactic gastrectomy specimens) to advanced GC in *CDH1* mutation carriers. However, variants of infectious agents such as aggressive strains of Helicobacter pylori and/or other environmental factors might be associated with this progression.

The c.88C>A (p.Pro30Thr) missense sequence variant was found in a family pedigree with a history of GC. Of note, this variant is rare in population databases (MAF 0.009 according to the ExAC consortium and 0.0004 according to 1000 genomes; available online https://www.ncbi.nlm.nih.gov/projects/SNP/snp_ref.cgi?do_not_redirect&rs=rs139866691). Notably, germline *CDH1* c.88C>A has been reported in lobular breast carcinoma and diffuse-type GC patients, as well as in two unrelated individuals with cleft lip with or without cleft palate, a developmental birth defect that is known to be overrepresented in *CDH1* mutation carriers [31,32,33,34]. It is often difficult to evaluate the pathogenicity of missense variants [14,22]. In silico prediction tools show conflicting interpretations when dealing with the *CDH1* c.88C>A variant pathogenicity [31]. In addition, c.88C>A has not been definitively classified by LOVD [10]. Of note, ClinVar has recently classified this variant as benign. 

The original criteria for HDGC outlined by the International Gastric Cancer Linkage Consortium IGCLC [6] established an initial basis for molecular testing. However, these criteria were significantly modified over time [5,8,9,12,35]. We used an age limit of 50 years of age, as previously used by Brooks-Wilson et al [35], because the aim of this study was to screen for the presence of *CDH1* variants in a population in which only few studies have been performed [18,19,36].

Previous studies have reported 14% to 50% of *CDH1* pathogenic sequence variants in patients who met clinical criteria for HDGC [8,9,10,16]. This study involved a rather low percentage of patients with clear pathogenic variants compared to other publications [8,9,10,16]. Important geographic differences have been described for GC with a low rate of *CDH1* germline mutations in high-risk areas, such as Japan and Korea [5,13]. Conversely, countries with a low rate of GC, such as New Zealand, have shown high rate of *CDH1* germline mutations, particularly among indigenous populations [31]. In contrast, our results were consistent with the expected low frequency of *CDH1* germline mutation rates observed in countries with high incidence and mortality rates, such as Chile [37]. The low number of pathogenic sequence variants in our study may be related to the difficulty in identifying true HDGC families using clinical criteria, due to the frequent occurrence of familiar clustering caused by non-genetic effects. Thus, further studies in Chile and other Latin American countries could be an interesting opportunity to study newly described genes in HDGC with differing mortality rates for GC [13,38]. Our group, as a member of The Latin American Gastric Cancer Genetics Collaborative Group, recently described the role of germline mutations in homologous recombination pathway related genes such as *PALB2*, *BRCA1*, and *RAD51C* in patients with GC [39]. Other groups have described other genes involved, such as *CTNNA1*, *ATM*, *ATR*, *FLCN*, *SBDS*, *MSR1*, *SDHB*, and *MAP3K6* [40], and genes altered in other genetic disorders [41]. These findings support the use of sequencing panels in the case of *CDH1*-mutation-negative patients. 

One of the limitations of this study is the inclusion criteria used, which do not exactly match the latest IGCLC guidelines [8], as these had not been described at the beginning of this study. If we applied the criteria of these guidelines, only 27 patients would have been included. Two of these patients (7.4%) would have been identified as carrying non-synonymous germline sequence variants, one of which (3.7%) would have been classified as pathogenic. Notably, there were no non-synonymous *CDH1* sequence variants identified in cases outside the 2015 IGCLC criteria. A second limitation of the study was that we did not perform multiplex ligation-dependent probe amplification (MLPA) analysis, which may detect large genomic rearrangements (LGRs) in high-risk families who are mutation-negative by sequencing approaches [42,43]. In addition, self-reported family history with lack of pathologic confirmation for relatives as described by other authors [5,35] may have limited the accuracy of the inclusion criteria applied. 

In summary, this prospective study associates the pathogenic variant c.1531C>T with the HDGC syndrome. Initial clinical features of this mutation are the early onset of disease. A laparoscopic total gastrectomy approach may improve the acceptance of a surgical option by *CDH1* mutation carriers. The low number of pathogenic variants found in this study could be an opportunity to study newly described genes in HDGC.

## 4. Materials and Methods 

In this prospective cross-sectional genetic study, we included GC patients over 18 of age with a biopsy-proven adenocarcinoma. Patients were evaluated at two affiliated hospitals, Hospital Sotero del Rio and Hospital Clinico Pontificia Universidad Católica de Chile from 2014 to 2016. The inclusion criteria for this study aimed to encompass diffuse-type GC cases with an early onset and/or family history of HDGC and were adapted from previous studies and the IGCLC 2010 guidelines [9,35]. The flowchart of the study is shown in Appendix A. The ethics committees at both participating hospitals approved this study; Comite Etico Cientifico Pontificia Universidad Catolica de Chile, Proyect number 14-044 (date 03/04/2014) and Comite de Evaluacion Etico-Cientifico Servicio de Salud Metropolitano Sur Oriente, Project number 2886 (date 08/14/2014). The study was conducted in accordance with the rules of the Helsinki declaration. Informed consent was obtained from all participants.

We assessed the probands’ demographic data, tumor pathology, clinical staging and family history. A dedicated clinical research nurse applied a standardized family history questionnaire. The questionnaire included age, sex and history of diffuse-type GC and other tumors in first, second and third-degree family members. A standardized pedigree was constructed according to recommendations of the National Society of Genetic Counselors [44]. The staging was based on the 7th edition of TNM-AJCC [45] and tumors were classified with Lauren’s criteria [46], from the surgical specimens (*n* = 26) or clinical evaluation based on endoscopic findings, biopsy and computed tomography imaging (*n* = 10).

### 4.1. DNA Extraction

Patients’ peripheral blood samples were collected in EDTA tubes. The buffy coat layer was isolated by centrifugation at 1800 rpm, for 20 min at 4 °C. Genomic DNA was extracted from the buffy coat using a QiAamp DNA Blood mini kit (Qiagen, Hilden, Germany) according to manufacturer’s instructions. All biological materials were blood DNA. All genetic analysis of the *CDH1* gene was on germline changes, not somatic.

### 4.2. Sequence Analysis

Analysis of the entire *CDH1* coding sequence, including the intron-exon boundaries and promoter region, was performed using a combination of Sanger sequencing and next-generation sequencing. Duel-indexed amplicon sequencing libraries for germline *CDH1* were generated using a two-step PCR strategy. Briefly, in the first PCR step, the coding exons of *CDH1* including their intron-exon borders and the proximal promoter were amplified. *CDH1* primers were designed with an additional known non-specific sequence that was used as a priming site for the second reaction. PCR products from the same study participant were pooled in equal volumes and purified using AMPure XP beads. In the second PCR step, pooled PCR products were amplified using a unique pair of indexed primers designed to add sequences necessary for multiplex sequencing on an Illumina MiSeq. To enable timely reporting of pathogenic variants, sample-specific libraries were pooled and sequenced in batches across multiple MiSeq runs. Amplification and sequencing processes have been previously described [31]. Raw paired-end reads were cleaned with Trimmomatic v.0.35. Cleaned reads were aligned with the human reference genome (GRCh37/hg19) using the Burrows-Wheeler Aligner v.0.7.10. Amplicons with low coverage (<40 reads) were sequenced again in a subsequent MiSeq run or Sanger sequenced. Variants were called using The Genome Analysis Toolkit’ (GATK) v.3.6. The effects of variants were predicted using SnpEff v.4.2.

*CDH1* sequence variants identified were detailed using recommendations from the Human Genome Variation Society [47], and interpretation of sequence variants was based on the American College of Medical Genetics and Genomics guidelines [22]. Non-synonymous variants were validated using Sanger sequencing.

In the families of probands with a pathogenic *CDH1* variant, we offered first and second-degree family members a clinical evaluation and *CDH1* sequencing of the variant identified in the proband. In patients with a confirmed pathogenic germline *CDH1* variant, we recommended an upper gastrointestinal endoscopy using narrow band imaging and biopsies of suspicious lesions as well as random gastric biopsies. Patients were counseled on the current guidelines for treatment options for carriers of pathogenic *CDH1* sequence variants and were referred to a multidisciplinary team composed of a clinical geneticist, an expert endoscopist and a gastric cancer surgery specialist [8].

### 4.3. Statistical Analyses

Statistical analyses were performed using SPSS version 22, Inc., Chicago, IL, USA. Categorical variables are expressed in percentages (%); quantitative values are expressed as the median (range).

## 5. Conclusions

This prospective hereditary gastric cancer study has identified a non-synonymous c.1531C>T pathogenic variant in the *CDH1* gene as a novel germline mutation of hereditary diffuse gastric cancer. Six family members carry the same nonsense pathogenic variant. Laparoscopic prophylactic total gastrectomy in the proband’s sister revealed stage I signet-ring cell carcinoma. This approach may improve the acceptance of a surgical option by *CDH1* mutation carriers. These findings warrant further study to uncover familial clustering of disease in *CDH1* negative patients. 

## Figures and Tables

**Figure 1 ijms-20-04980-f001:**
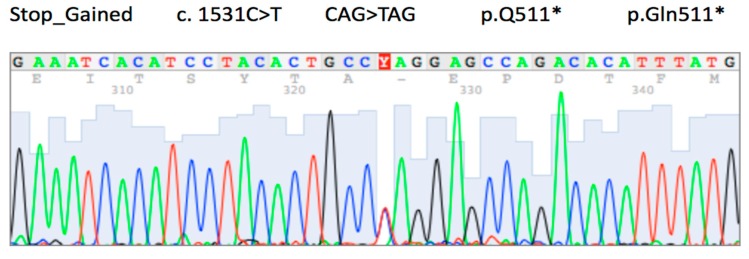
Partial Electropherogram showing the C>T change in the proband identified with the c.1531C>T pathogenic mutation. The red Y shows the place in the sequence where a double peak occurs (heterozygous Genotype). * Incomplete protein.

**Figure 2 ijms-20-04980-f002:**
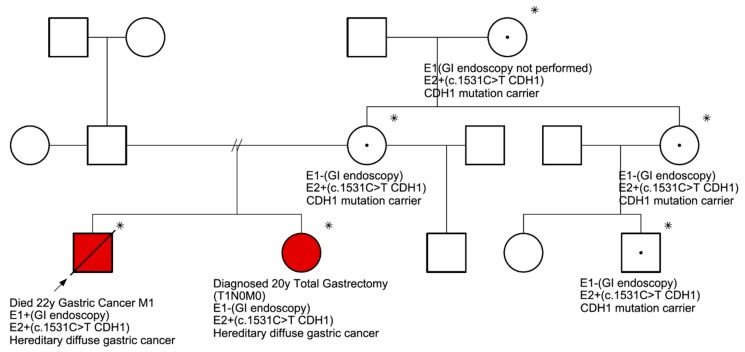
Family pedigree with c.1531C>T (p.Gln511*) nonsense sequence variant in six family members showing hereditary diffuse gastric cancer (HDGC) syndrome. * Patients with c.1531C>T sequence variant. 
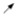
: Proband. 
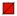
: Died from gastric cancer. 
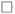
: Male. 
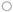
: Female. E1: endoscopic evaluation. E2: molecular evaluation. In red, the two individuals in which DGC was confirmed.

**Figure 3 ijms-20-04980-f003:**
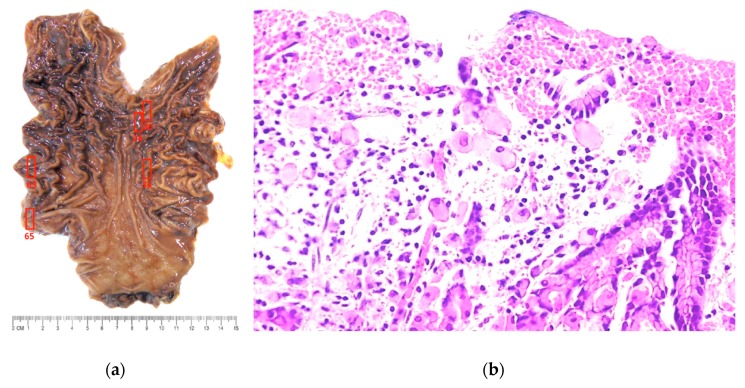
(**a**) Prophylactic gastrectomy specimen with no visible macroscopic abnormalities. Complete mapping revealed five regions where microscopic foci of intramucosal signet ring cell carcinoma was found (red boxes). (**b**) Photomicrograph shows foci of signet-ring cell carcinoma, involving the gastric mucosa (stage IA disease) (Hematoxilin-eosin stain, 100X).

**Table 1 ijms-20-04980-t001:** Clinical characteristics of proband included in the study.

Characteristics	N (%)
**Total**	36
**Gender**	
Male	18 (50)
Female	18 (50)
**Age (years)**	44 (range: 22–68)
≤ 29	7 (19)
30–39	7 (19)
40–49	12 (33)
50–59	4 (11)
60 or older	6 (16)
**Smoking status**	
Current smoker	16 (44)
Previously smoker	3 (8)
Non smoker	17 (47)
**Stage**	
I	5 (14)
II	6 (17)
III	12 (33)
IV	13 (36)
**Family history of gastric cancer ***	
1 family member (only proband)	17 (47)
2 family members	6 (17)
3 family members	9 (25)
4 or more family members	4 (11)
Family history of breast cancer	6 (16)
Family history of colorectal cancer	2 (6)
Family history of prostate cancer	3 (8)
Family history of uterine cancer	2 (6)
Family history of skin cancer	1 (3)
Family history of esophageal cancer	1 (3)
Family history of gallbladder cancer	1 (3)
Family history of lung cancer	1 (3)
Family history of thyroid cancer	1 (3)
Family history of ovarian cancer	1 (3)
Family history of other cancer unknown	6 (16)
**Helicobacter pylori infection**	
Infected	15 (42)
Not infected	10 (28)
Unknown	11 (30)

* Including proband and first and second-degree family members.

**Table 2 ijms-20-04980-t002:** *CDH1* variants identified and pathologic significance.

SNP ID	Sequence Variant	Protein Change	Probands (n)	Location	Class	Significance ^α^
rs16260	c.-285C>A		19	Promoter	Promoter	Non-coding
rs28372783	c.-197A>C		11	Promoter	Promoter	Non-coding
rs3743674	c.48+6C>T		36	Intron 1	Splice site	Benign
rs139866691	c.88C>A	p.Pro30Thr	1	Exon 2	Missense	Benign
rs33963999	c.531+10G>C		1	Intron 4	Splice site	Benign
rs61756284	c.1272A>T	p.Thr424Thr	1	Exon 9	Synonymous	Benign
rs1131690810	c.1531C>T	p.Gln511 *	1	Exon 10	Nonsense	Pathogenic
rs786201452	c.1893A>T	p.Thr631Thr	1	Exon 12	Synonymous	Likely benign
rs764379691	c.2052C>T	p.Ser684Ser	1	Exon 13	Synonymous	Likely benign
rs1801552	c.2076T>C	p.Ala692Ala	30	Exon 13	Synonymous	Benign
rs33964119	c.2253C>T	p.Asn751Asn	4	Exon 14	Synonymous	Benign

^α^ Significance according to ClinVar [20]. * Incomplete protein.

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
