# Peer review of "Identification of c.1531C>T Pathogenic Variant in the CDH1 Gene as a Novel Germline Mutation of Hereditary Diffuse Gastric Cancer"

_ijms, 2019, doi:10.3390/ijms20204980_

Round 1
Reviewer 1 Report
This paper is a revised manuscript, earlier reviewed as ijms-484039, describing pathogenic germline variants (including a novel one) of the CDH1 gene, in Chilean patients with diffuse gastric cancer. The study will be valuable for clinical geneticists.
Some textual problems remain, either introduced during revision or overlooked in the first review.
In the abstract, line 26, the text beginning “a multidisciplinary team” is unintelligible. Do the authors mean “from genomic DNA, and a multidisciplinary team managed each family member with a pathogenic sequence variant”?
Line 32: Please correct to “pathogenic variant was identified”.
Line 41: “1.033.000” should be “1,033,000”.
Line 57: Please add a comma after “cell adhesion”.
Line 70: Presumably the authors mean “family members with DGC diagnosed at any age”.
Line 77: The repetition of the breast cancer and colorectal cancer numbers is not necessary because they are in Table 1. The added text “in the patient families included” would be better as “found in the patient families”.
Line 147, Figure 3 legend: Please correct to “five regions where microscopic foci”.
Line 148: Please correct to “Photomicrograph shows”.
Line 163: Please correct to “review reports that the final histopathology diagnosis of GC was made in 87%”.
Line 166: This text is confusing. Do the authors mean “total gastrectomy in our study and in reports from other groups”? What is intended about the other groups is not clear.
Line 169: Please correct to “older family members did not”.
Line 201: Please correct to “GC with a low rate”.
Line 219: Please correct to “the criteria of these guidelines”.
Line 231: Please correct to “A laparoscopic total gastrectomy”.
Line 235: Please correct to “biopsy-proven adenocarcinoma”.
It is not clear why the sections entitled “Appendix A” and “Appendix B” are present in the reviewer’s copy, as these seem to be instructions to the authors.
In the references, most journal titles with multi-word names are inappropriately capitalized.
Author Response
Response to reviewer:
This paper is a revised manuscript, earlier reviewed as ijms-484039, describing pathogenic germline variants (including a novel one) of the CDH1 gene, in Chilean patients with diffuse gastric cancer. The study will be valuable for clinical geneticists.
Some textual problems remain, either introduced during revision or overlooked in the first review.
In the abstract, line 26, the text beginning “a multidisciplinary team” is unintelligible. Do the authors mean “from genomic DNA, and a multidisciplinary team managed each family member with a pathogenic sequence variant”?
Corrected in the text
Line 32: Please correct to “pathogenic variant was identified”.
Corrected in the text
Line 41: “1.033.000” should be “1,033,000”.
Corrected in the text
Line 57: Please add a comma after “cell adhesion”.
Corrected in the text
Line 70: Presumably the authors mean “family members with DGC diagnosed at any age”.
Corrected in the text
Line 77: The repetition of the breast cancer and colorectal cancer numbers is not necessary because they are in Table 1. The added text “in the patient families included” would be better as “found in the patient families”.
Corrected in the text
Line 147, Figure 3 legend: Please correct to “five regions where microscopic foci”.
Corrected in the text
Line 148: Please correct to “Photomicrograph shows”.
Corrected in the text
Line 163: Please correct to “review reports that the final histopathology diagnosis of GC was made in 87%”.
Corrected in the text
Line 166: This text is confusing. Do the authors mean “total gastrectomy in our study and in reports from other groups”? What is intended about the other groups is not clear.
Corrected in the text
Line 169: Please correct to “older family members did not”.
Corrected in the text
Line 201: Please correct to “GC with a low rate”.
Corrected in the text
Line 219: Please correct to “the criteria of these guidelines”.
Corrected in the text
Line 231: Please correct to “A laparoscopic total gastrectomy”.
Corrected in the text
Line 235: Please correct to “biopsy-proven adenocarcinoma”.
Corrected in the text
It is not clear why the sections entitled “Appendix A” and “Appendix B” are present in the reviewer’s copy, as these seem to be instructions to the authors.
Corrected in the text, this was erased
In the references, most journal titles with multi-word names are inappropriately capitalized.
We corrected this in the text in the references section
Reviewer 2 Report
Authors correcly answer to the requests
Author Response
OK
This manuscript is a resubmission of an earlier submission. The following is a list of the peer review reports and author responses from that submission.
Round 1
Reviewer 1 Report
The study is complete and well perfomed. It is of particualr interst for Chile since only in the last years data of CDH1 mutation have been studied. The report of new mutation wthin well characterised family is important for all the scientific community interested on hereditary form of tumors.
Mutation abolish the E-cadherin exopression in the tumor tissue ?? Please added the immunohistochemical data to demonstrated the potential function of the mutation in vivo.
Please specificy the type of tumors reported as "other" in table 1
To complete the information of cases You must specify the criteria of inclusion for cases with only one family member with GC.
same question : 26 patients had only one criteria of inclusion, 19 had at least 2 family members with GC the other ? Please specify
Have data regarding Helicobcater pylori infection ? or smking ? 4 members of the family with the same mutations altough older have no developed GC, Could you propose a reason for this in the discussion (altough speculative).
Added some sentences regarding the association o fthe same mutation with other hereditary diseases
Author Response
Reviewer 1.
The study is complete and well perfomed. It is of particualr interst for Chile since only in the last years data of CDH1 mutation have been studied. The report of new mutation wthin well characterised family is important for all the scientific community interested on hereditary form of tumors.
Mutation abolish the E-cadherin exopression in the tumor tissue ?? Please added the immunohistochemical data to demonstrated the potential function of the mutation in vivo.
Response:
We believe that it would be very interesting to perform inmunohistochemistry in the tumor and to evaluate e-cadherin expression as stated by the reviewer. We tried to do this in the microfoci of the prophilactic total gastrectomy specimen, but unfortunatedly the foci were to small to evaluate this accurately.
Please specificy the type of tumors reported as "other" in table 1
Response:
We added the other tumors in table 1
Family history of gastric cancer | |
1 family member (only proband) | 17 (47) |
2 family members | 6 (17) |
3 family members | 9 (25) |
4 or more family members | 4 (11) |
Family history of breast cancer | 6 (16) |
Family history of colorectal cancer | 2 (6) |
Family history of prostate cancer | 2 (6) |
Family history of uterine cancer | 2 (6) |
Family history of skin cancer | 1 (3) |
Family history of esophageal cancer | 1 (3) |
Family history of gallbladder cancer | 1 (3) |
Family history of lung cancer | 1 (3) |
Family history of thyroid cancer | 1 (3) |
Family history of ovarian cancer | 1 (3) |
Family history of other cancer unknown | 6 (16) |
To complete the information of cases You must specify the criteria of inclusion for cases with only one family member with GC.
same question : 26 patients had only one criteria of inclusion, 19 had at least 2 family members with GC the other ? Please specify
Response:
In the matherial and methods section we stated: “The inclusion criteria for this study aimed to encompass diffuse-type GC cases with an early onset and/or family history of HDGC and were adapted from previous studies and the IGCLC 2010 guidelines [9,35]. The Flowchart of the study is shown in sFigure. 2.” And inclusion criteria are described:
-Family history of DGC in first or second-degree family members, with one ≤ 50 years
-Three or more family members with DGC of any age
-Patients with the diagnosis of DGC ≤ 50 years
In addition, in the We re-wrote this in the results section:
We erased “Overall, 24 (66%) patients met one inclusion criteria, 6 (16%) met two and 6 (16%) families met all three inclusion criteria”
We erased: “In 17 (47%) patients, the proband was the only family member with diffuse-type GC. The remaining 19 (53%) patients had first- or second-degree family members who had been diagnosed with diffuse-type GC (Table 1). In this latter group, there was a mean of three (two to six) relatives with diffuse-type GC”.
Instead we added: There were 13 patients with family history of DGC in first or second-degree family members, with one ≤ 50 years; 13 probands had three or more family members with DGC of any age; and 27 patients had a diagnosis of DGC ≤ 50 years. Seventeen patients (47%) had no other family members with a diagnosis of GC, and they were included in the study because they had age ≤ 50 years when they were diagnosed with GC
Have data regarding Helicobcater pylori infection ? or smking ? 4 members of the family with the same mutations altough older have no developed GC, Could you propose a reason for this in the discussion (altough speculative).
Response:
We have data about Helicobacter pylori infection in 25 patients; for the other 11 patients helicobacter pilory infection was unknown. We know helicobacter pylory infection was present en 15 / 25 patients.
We have data on smoking status; 16 patients were active smokers, 3 had smoked in the past but stopped and 17 patients were not smokers.
We added this in table 1:
Smoking status | |
Current smoker | 16 (44) |
Previously smoker | 3 (8) |
Non smoker | 17 (47) |
Helicobacter pylori infection | |
Infected | 15 (42) |
Not infected | 10 (28) |
Unknown | 11 (30) |
Response to development of advanced GC in the younger, but not in the older family members:
We added this to the discussion section:
The family described in our study with the c.1531C>T variant has a unique profile, in which one of the youngest family members developed an advanced GC and older family members do not. We do not understand the full mechanisms behind the progression from normal mucosa or small foci of signet ring cells (present in most prophylactic gastrectomy specimens) to advanced GC in CDH1 mutation carriers. However, variants of infectious agents such as aggressive strains of helicobacter pylori or other environmental factors might be associated with this progression.
Added some sentences regarding the association o fthe same mutation with other hereditary diseases
To our knowledge this prospective genetic study is the first to describe the pathogenic variant c.1531C>T, in association with any hereditary syndromes.
We deleted this sentence: “Although this mutation has been linked to a wide range of hereditary cancer-predisposing syndromes (MedGen: C0027672) its role in HDGC remains to be defined”
Reviewer 2 Report
This manuscript reports a germline variant in CDH1 in a Chilean family with diffuse gastric cancer. A prophylactic gastrectomy in an asymptomatic carrier showed stage 1A diffuse gastric cancer. The study will be useful for clinical geneticists, and will likely inform families with clusters of cancers. Chile has a high incidence of gastric cancer, and thus has the patient population to examine such genetic variants.
The paper is in general well-written, but has a few places where changes can be made for correctness or clarity.
Line 64: Please correct to “in the CDH1 gene”.
The sentence beginning on line 69 is awkward and indicates that the variant was the person. It could be edited as “The first proband, found to carry a missense c.88C>A (p.Pro30Thr, rs139866691) variant, was a 59-year- old female diagnosed with stage III GC of signet-ring cell subtype”.
Line 83: Please correct to “The second proband, belonging to a different family, was found”. The missing commas change the meaning from the one intended.
Line 148: Do the authors mean “in which only a few studies have been performed”?
Line 153: Please correct to “with a low rate”.
Line 154: It is not clear what is meant by “In this scenario”. In indigenous populations? If the authors mean to contrast to the New Zealand studies, they could substitute “in this scenario” by “In contrast” or “In our study”.
Line 166: Please correct to “CDH1-mutation-negative patients”.
Line 167: Please correct to “which do not exactly”, because “criteria” is plural.
Line 169: Please correct to “these guidelines’ criteria”, or better, because less awkward, “the criteria of these guidelines”.
Line 173: This sentence is confusing in several ways. It implies that the reason MLPA was not done was because it can detect large genomic rearrangements. Also the first part of the sentence is awkward, and uses “pendent” instead of “dependent”. One way to improve could be “A second limitation of the study was that we did not perform multiplex ligation-dependent probe amplification (MLPA) analysis, which may detect large genomic rearrangements (LGRs) in high-risk families who are mutation-negative by sequencing approaches”.
Line 190: Please correct to “the probands’ demographic data’.
Line 195: This sentence is confusing. It might be improved as “Lauren’s criteria, from the surgical specimens”.
Line 200: A degree symbol is missing.
Line 214: Please correct to “sample-specific”.
Line 221: Do the authors refer to the one sequence variant that is the focus of this paper, or to all the ones sequenced in the study? If only one, please correct to “The CDH1 sequence variant identified was detailed” and “interpretation of the sequence variant identified” in Line 222. If the authors mean all the sequences, then please correct to “The CDH1 sequence variants identified” and “interpretation of sequence variants”.
Line 228 has “asP” at the end of the line, presumably a typographical error.
Line 231: Please correct to “composed of a”.
Line 238: Please correct to “Laparoscopic prophylactic total gastrectomy”. An alternative is to leave word order as given, but surround “prophylactic in the proband’s sister” with commas.
Please use standard gene name conventions (italics) for all gene names. This is not done in the Supplement.
Author Response
Reviewer II
This manuscript reports a germline variant in CDH1 in a Chilean family with diffuse gastric cancer. A prophylactic gastrectomy in an asymptomatic carrier showed stage 1A diffuse gastric cancer. The study will be useful for clinical geneticists, and will likely inform families with clusters of cancers. Chile has a high incidence of gastric cancer, and thus has the patient population to examine such genetic variants.
The paper is in general well-written, but has a few places where changes can be made for correctness or clarity.
Line 64: Please correct to “in the CDH1 gene”.
Response: Corrected in the text.
The sentence beginning on line 69 is awkward and indicates that the variant was the person. It could be edited as “The first proband, found to carry a missense c.88C>A (p.Pro30Thr, rs139866691) variant, was a 59-year- old female diagnosed with stage III GC of signet-ring cell subtype”.
Response: Corrected in the text.
Line 83: Please correct to “The second proband, belonging to a different family, was found”. The missing commas change the meaning from the one intended.
Response: Corrected in the text.
Line 148: Do the authors mean “in which only a few studies have been performed”?
Response: Corrected in the text.
Line 153: Please correct to “with a low rate”.
Response: Corrected in the text.
Line 154: It is not clear what is meant by “In this scenario”. In indigenous populations? If the authors mean to contrast to the New Zealand studies, they could substitute “in this scenario” by “In contrast” or “In our study”.
Response: Corrected in the text.
Line 166: Please correct to “CDH1-mutation-negative patients”.
Response: Corrected in the text.
Line 167: Please correct to “which do not exactly”, because “criteria” is plural.
Response: Corrected in the text.
Line 169: Please correct to “these guidelines’ criteria”, or better, because less awkward, “the criteria of these guidelines”.
Response: Corrected in the text.
Line 173: This sentence is confusing in several ways. It implies that the reason MLPA was not done was because it can detect large genomic rearrangements. Also the first part of the sentence is awkward, and uses “pendent” instead of “dependent”. One way to improve could be “A second limitation of the study was that we did not perform multiplex ligation-dependent probe amplification (MLPA) analysis, which may detect large genomic rearrangements (LGRs) in high-risk families who are mutation-negative by sequencing approaches”.
Response: Corrected in the text.
Line 190: Please correct to “the probands’ demographic data’.
Response: Corrected in the text.
Line 195: This sentence is confusing. It might be improved as “Lauren’s criteria, from the surgical specimens”.
Response: Corrected in the text.
Line 200: A degree symbol is missing.
Response: Corrected in the text.
Line 214: Please correct to “sample-specific”.
Response: Corrected in the text.
Line 221: Do the authors refer to the one sequence variant that is the focus of this paper, or to all the ones sequenced in the study? If only one, please correct to “The CDH1 sequence variant identified was detailed” and “interpretation of the sequence variant identified” in Line 222. If the authors mean all the sequences, then please correct to “The CDH1 sequence variants identified” and “interpretation of sequence variants”.
Response: All sequences, Corrected in the text.
Line 228 has “asP” at the end of the line, presumably a typographical error.
Response: Corrected in the text.
Line 231: Please correct to “composed of a”.
Response: Corrected in the text.
Line 238: Please correct to “Laparoscopic prophylactic total gastrectomy”. An alternative is to leave word order as given, but surround “prophylactic in the proband’s sister” with commas.
Response: Corrected in the text.
Please use standard gene name conventions (italics) for all gene names. This is not done in the Supplement.
Response: Corrected in the text.